# Unlocking the Full Potential of Data Science Requires Tabular Foundation Models, Agents, and Humans

## Abstract

Despite its vast potential, data science remains constrained by manual workflows and fragmented tools. Meanwhile, foundation models have transformed natural language and computer vision — and are beginning to bring similar breakthroughs to structured data, particularly the ubiquitous tabular data central to data science. At the same time, there are strong claims that fully autonomous agentic data science systems will emerge. We argue that, rather than replacing data scientists, the future of data science lies in a new paradigm that amplifies their impact: collaborative systems that tightly integrate agents and tabular foundation models (TFMs) with human experts. In this paper, we discuss the potential and challenges of navigating the interplay between these three and present a research agenda to guide this disruption toward a more accessible, robust, and human-centered data science.

## 1 The Disruption of Data Science is Inevitable

Over the past two decades, data science has accelerated and improved decision-making. Fueled by increased data availability, advances in machine learning, easy-to-use software packages, and scalable compute infrastructure, data science has transformed applications across various domains. Yet, data science has not reached its full potential, especially for tabular data. Workflows remain fragmented across a diversity of tools and labor-intensive [114, 101], and the gap between technical experts and domain stakeholders continues to limit mutual understanding [112].

In recent years, foundation models have revolutionized text and images, and are starting to do the same in structured domains, such as tabular data. Tabular Foundation Models (TFMs), such as TabPFN [61] and TableGPT2 [124], demonstrate that models pretrained across heterogeneous tables can generalize well to new tables for predictive tasks, data wrangling, and beyond. Simultaneously, the rise of autonomous agentic systems capable of reasoning, tool use, and coding offers a perfect match to the many repetitive and procedural tasks that dominate tabular data science workflows. Together, foundation models and agents clearly herald a major shift for tabular data science as they have the potential to automate data science end-to-end, which is very different from prior, more siloed attempts such as AutoML. However, instead of replacing human experts by automation, we argue that the future of data science lies in a new paradigm that amplifies the impact of human experts:

> **Our Position**
>
> **Owing to rapid developments in agentic data science and tabular foundation models, the role of data scientists is set to undergo massive disruption. Instead of chasing the "AI replacing data scientists" scenario, we argue that tabular foundation models, agents, and human experts, together, hold the key to unlocking the full potential of data science, if the risks and opportunities inherent to their interplay are navigated effectively.**

While it is clear that agents and TFMs will continue to advance rapidly, this position paper argues that human involvement remains indispensable. Crucially, humans are often the sole source of institutional knowledge—knowledge that is deeply contextual, environment-specific, and typically undocumented and thus not accessible to either TFMs or agents. Furthermore, humans are essential for disambiguating edge cases, resolving blind spots in model reasoning, and serving as the final arbiters of correctness. They provide critical judgment in determining whether the outputs of automated systems are valid and trustworthy.

Although the concept of keeping humans in the loop is not new, we advocate for a *more radical approach*—one in which agents, TFMs, and humans are brought together as tightly coupled collaborators. When designed thoughtfully, this triad can complement and amplify each other's strengths, driving a new era of accelerated, robust data science. The central challenge we pose in this paper is how to orchestrate the collaboration, since navigating the interaction between agents, TFMs, and humans is far from trivial—especially as the capabilities of each component continue to evolve.

## 2  Tabular Data Science has Not Reached its Full Potential

Data science has matured substantially over the last 50 years, driving significant advancements across diverse sectors, including use cases such as enhanced diagnostic precision in healthcare [9, 76, 4], recommendation systems in retail [83, 37, 89, 14], and improved investment performance in finance [28, 55, 30] amongst many others. Despite its significance and widespread use, data science, however, has by far not yet reached its full potential, with a substantial gap persisting between current practices and the ultimate potential it could have for businesses and society.

### 2.1  Challenges to Data Science

Data science projects often fall short of their promised value because of recurring challenges.

**The Sobering Reality: High Failure Rates.** Recent analyses consistently indicate that a significant portion of data science and AI initiatives encounter substantial hurdles, often falling short of their intended business objectives or failing to achieve full production deployment [114, 131, 69, 134]. These considerable failure rate highlights systemic challenges prevalent throughout the data science lifecycle. These difficulties primarily originate from two critical areas: data-related issues and model-related issues [112, 135].

**Under-utilized & Fragmented Data.** A significant volume of data, often called *dark data* [16], remains under-utilized because of limited resources to navigate data, inadequate tools, or a lack of awareness regarding its inherent value. This widespread under-utilization of data directly contributes to suboptimal decisions, as countless business choices continue to be made based on incomplete information, rather than leveraging data-driven insights that could yield superior outcomes, cost savings, or new revenue streams. Furthermore, persistent integration challenges arise from data accessibility issues and data silos, with data locked in disparate systems [77, 119, 35, 3, 113, 88] as well as the lack of standardized definitions [121, 133].

**Data Preparation as Impediment.** Data preparation (collecting, integrating, cleaning, and transforming data) consumes a significant portion, approximately 80%, of a data scientist's time, leaving only 20% for actual analysis and model building [24, 119, 112, 114]. Despite long-standing efforts to address data quality and cleaning [105, 52, 34, 119, 65, 42], these remain substantial hurdles in practice. Overall, this leads to a diversion of highly skilled data scientists to often tedious and repetitive data work, which impedes high-impact tasks and responsibilities like model development and interpretation, leading to project delays and missed business opportunities.

**Model Failures.** Data scientists make use of models to capture the patterns in data, which can go wrong in many ways. Inadequate feature engineering–the process of selecting, transforming, and creating model inputs–can severely limit model performance. Other common pitfalls like over- or underfitting can compromise model generalizability [51, 142, 127]. Beyond mere prediction performance, the validity of conclusions may be compromised by unaccounted-for confounders [1]. In addition to these, the misapplication of techniques often results in the selection of inappropriate or overly complex algorithms, undermining project success [32, 11]. But even if that succeeds, models used in dynamic environments degrade over time due to concept and data drift [38]. In all these cases, unnecessary model complexity creates technical debt [114].

Table 1: Complementary strengths of human experts, alongside TFMs and LLM Agents. Only their combination covers the full spectrum of capabilities needed for agentic data science.

| Capability | Humans | | TFMs | LLM Agents |
| --- | --- | --- | --- | --- |
| | Data Scientists | Domain Experts | | |
| Domain knowledge understanding | ✗ | ✓ | ✗ | ✗ |
| Tabular structure understanding | ✓ | ✗ | ✓ | ✗ |
| Contextual data science | ✓ | ✗ | ✗ | ✗ |
| Planning & ML tool execution | ✓ | ✗ | ✗ | ✓ |
| Goal alignment | ✗ | ✓ | ✗ | ✗ |
| Scalable automation | ✗ | ✗ | ✓ | ✓ |

## 2.2 Envisioning the Full Potential of Data Science

Many of the aforementioned challenges stem from the limited bandwidth of data scientists and the highly manual, repetitive work required to navigate the vast space of possible solutions, like selecting appropriate datasets and determining how to combine models with data. In recent years, numerous efforts have thus aimed to address these issues and to automate various data-related tasks, including data exploration, transformation, and cleaning. Similarly, in model construction, tools like AutoML have sought to automate the design and tuning of machine learning models. But mere automation, while reducing overhead, does not unlock the full potential of data science. It risks reinforcing existing patterns while only marginally expanding the solution space. To truly innovate, we need systems that can creatively and effectively explore novel solutions. We believe that a tightly integrated combination of agents, TFMs, and human experts can enable the kind of exploration and creativity needed to catalyze data science. Moreover, such a system should empower a broader spectrum of users, from domain experts to data scientists, who engage with data systems in different ways to extract insights. The foundational building blocks are now in place to begin making it a reality.

## 3 TFMs and LLM Agents Fall Short in Isolation, and without Humans

Recent attempts to supercharge data science have focused primarily on automating individual steps of the data science workflow, with two promising directions emerging. LLM-based agents and their table-tuned variants [79, 123] can perform a wide range of tasks directly on tables but often lack a deep, structural understanding of tabular semantics and statistical reasoning. Conversely, the current generation of tabular foundation models [61, 103, 27, 59] better capture table structure but fall short in flexibility and task coverage. However, while automation helps to scale data science, we also show why human input remains indispensable and cannot be fully replaced. Table 1 summarizes the capabilities and limitations of LLM agents, TFMs, and human experts, serving as a reference throughout the discussion.

### 3.1 TFMs: Understand Tabular Structures, but Limited Capabilities

TFMs are purpose-built architectures designed to tackle tasks on tabular data out of the box or with minimal overhead. Their structure enables them to better understand table-specific patterns and relationships. Presently, TFMs typically fall into two categories:

- **Predictive TFMs** like TabPFN [60, 61], TabICL [103], CARTE [72], and other variants focus on predictive machine learning tasks like classification and regression.
- **Representation TFMs** like TaBERT [139], TURL [27], and TaPas [59] produce task-agnostic embeddings for downstream models or fine-tuning for specific tasks such as column type prediction, table question answering, and entity matching [6].

**TFMs are Efficient but Limited.** Importantly, both classes of TFMs are tabular-native; they employ both row-wise and column-wise attention within transformer-based frameworks, and they often natively process mixed numerical and categorical data without costly byte-pair tokenization. Synthetic dataset pre-training further imparts invariance to row and column order, as well as robustness to missing-value patterns, at scale. CARTE [72] and TARTE [73] additionally incorporate column

names to bootstrap from knowledge graphs, and KumoRFM [106] also handles relational data. Moreover, TFMs are sample efficient in predictions and their architectures allow them to remain compact—typically within the 10–50M parameter range, which is two to four orders of magnitude smaller than frontier LLMs. As a result, they benefit from reduced inference latency, lower energy consumption, and a smaller carbon footprint [122, 66]. Nonetheless, TFMs lack many of the broader capabilities expected from true foundation models which are available in modern LLMs.

**Why Today's TFMs Are Insufficient.** The success of foundation models in other modalities stems from their broad task coverage, general-purpose representations, and ability to be adapted with minimal additional supervision [13, 2, 104]. In contrast, today's TFMs fall short of this ideal. Predictive TFMs are narrowly scoped to row-level classification or regression, addressing only a small fraction of the data science workflow, leaving many crucial tasks (e.g., data cleaning, which may require awareness of multiple rows or entire tables) unaddressed. Representation TFMs offer reusable embeddings across broader tasks, but they currently still require separate downstream models or training for each specific task [6]. This breaks the promise of end-to-end adaptability and adds complexity to deployment. As a result, today's TFMs do not yet match the versatility and integrative power that define true foundation models.

## 3.2 LLM Agents: Generalist Abilities, but Lacking Rigor

In contrast to TFMs, LLMs are generalists, excelling across a wide range of tasks. Their ability to follow natural language instructions and generalize from minimal examples based on their background knowledge makes them appealing as universal assistants throughout data science workflows. LLM *agents* further extend their capabilities to multi-step reasoning and tool use, enabling them to invoke external functions, APIs, or code—capabilities highly relevant for complex data science tasks. However, LLMs alone remain insufficient for tasks that require statistical reasoning, such as complex table understanding or prediction tasks.

**LLMs Lack Rigorous Table Understanding.** Although LLMs have achieved strong results on certain data wrangling [93] and exploration tasks (e.g., Text-to-SQL [118]), substantial evidence shows that they still lack a rigorous and reliable understanding of structured table data [22, 136]. While LLMs excel at predictive tasks with a handful of data points (due to their strong background knowledge) [56, 41], they are not capable of statistical reasoning for more than a few dozen data points. The limitations are especially pronounced in enterprise scenarios, where data is highly complex and domain-specific, and LLMs cause high computational costs when applied to large tables [12, 95]. Furthermore, LLMs' internal reasoning processes are opaque, making it difficult to interpret model outputs or systematically improve performance [50, 136].

**Why Reasoning and Tools Are Not Enough.** Enhancements such as retrieval-augmented generation and integrated tool use have been developed to address LLM challenges like hallucination and data unawareness. Yet, these methods remain limited. LLMs frequently hallucinate logic, struggle with genuinely complex multi-step reasoning, and often lack awareness of what data is needed to achieve specific analysis goals [19]. Although they may access multiple tools, LLMs generally lack a dedicated planner to coordinate and optimize tool invocation in complex action sequences effectively. As a result, their execution capabilities are restricted, particularly when facing open-ended or loosely defined objectives [40].

## 3.3 Human Experts: Limited Bandwidth, but Indispensable Partners

Human experts face clear challenges, including limited time, cognitive bandwidth, and difficulty in systematically exploring large and complex solution spaces. A study among AI practitioners [112], for example, highlighted that data preparation work is "time-consuming, invisible to track, and often done under pressures to move fast due to margins — investment, constraints, and deadlines often came in the way of focusing on improving data quality." Compromises on problem formulation, data quality, and critical reflection on modeling validity are inevitable leading to suboptimal data science outcomes. Nevertheless, humans bring expertise that remains critical for solving data science problems. This expertise broadly falls into two categories: domain expertise and data science expertise.

**Domain Expertise.** The knowledge to solve data-related problems is often only available in the institutional knowledge of teams and humans and is usually not documented and thus machine accessible at all. Domain experts are thus critically required to help resolve ambiguity, uncover

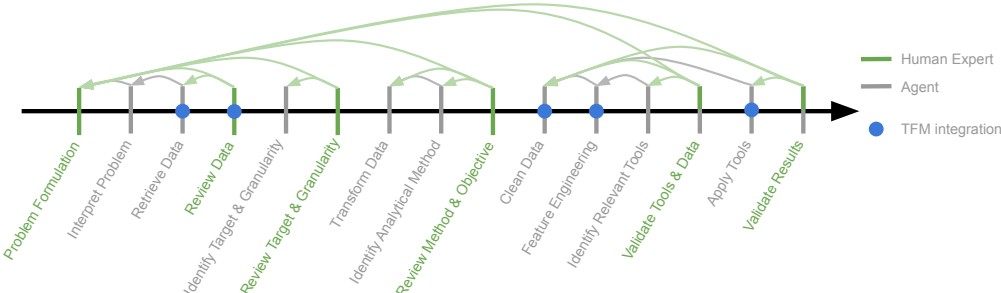

Figure 1: An example of what a data science workflow could look like. Different operations are executed by the human expert and agent, some with the help of tabular foundation models.

implicit assumptions, and adapt to evolving goals, tasks that rely on human insight, especially when important context is missing from the data. For example, only a local expert might know that in some radiology departments, but not in others, reports on follow-up scans omit previously-known findings [129]—a fact not encoded in structured data or documentation. Domain experts are also essential for interpreting outcomes, aligning analyses with undocumented clinical goals, or catching when something simply does not "feel right." These nuanced judgments rely on intuitions and tacit domain knowledge that current systems cannot capture [115].

**Data Science Expertise.** Contextual intelligence is crucial for effective and responsible data science. Data scientists are adept at framing the right problem given the organizational and societal context, selecting tools and methods that are both valid for the data at hand and appropriate for the task's objectives. Unlike automated systems, they can assess whether results make sense, especially in performance-critical or ethically sensitive settings [25]. For instance, a model predicting hospital readmissions may appear statistically accurate, but a data scientist might flag it as untrustworthy if it relies on biased care patterns, such as historically underserved patients receiving less follow-up, not because they were healthier, but due to systemic inequities. Moreover, they understand real-world deployment constraints, from latency requirements to interpretability, and can steer solutions accordingly. These capabilities do and should augment semi-automated systems [132].

## 4 Realizing the Potential of Data Science with Agents, TFMs, and Humans

LLM agents, TFMs, and human experts, when combined in tight collaboration, form a highly promising "architecture" for collaborative data science systems, covering a full spectrum of capabilities, shown in Table 1. Humans contribute intent, domain knowledge, and critical judgment for goal alignment. TFMs offer structure-aware understanding and statistical reasoning. Agents serve as orchestrators, able to plan and call tools and converse with humans. Any two without the other would leave critical gaps, be it a lack of alignment, understanding, or scalable automation of multi-step workflows. In the following, we discuss why and how to bring them together.

### 4.1 Synergies of Bringing Agents, TFMs, and Humans Together

Collaboration between human experts and TFM-equipped agents has the potential to elevate data science well beyond the capabilities of individual components. Collaborating with TFM-equipped agents can accelerate many time-consuming steps in data science. For instance, laborious data work can be augmented by agents leveraging specialized TFMs for data wrangling [39, 81, 93] or data analysis [98]. Predictive TFMs like TabPFN [60, 61] alleviate the need for extensive hyperparameter tuning and modeling work for predictive tasks. Thus, agents applying TFMs can elevate human roles, enabling data scientists to create more value and empowering domain experts to contribute directly.

**Unlocking Data Scientists for Critical Work.** With LLM agents and TFMs handling more repetitive work, data scientists can dedicate their efforts to higher-value activities. Their expertise remains crucial in framing the problem by identifying the right questions and requisite data [11], especially since agents can misinterpret task details, leading to incorrect results [63]. Oversight is also vital for

steering workflows, critically evaluating intermediate results, and ensuring alignment with project goals and real-world complexities. This is particularly important given agents' documented struggles with precise instruction adherence, task memory, and strategic planning in dynamic settings [63, 141]. The dynamic and uncertain nature of real-world analysis necessitates effective human-agent interaction, where data scientists navigate ambiguous user intents and adjust strategies based on intermediate findings [82]. Thus, the collaboration allows and requires data scientists to shift their focus towards scrutinizing the data science workflow through deeper critical thinking, robust and continuous validation, considering ethical consequences, and maintaining overall project integrity, which are all factors observed as major determinants of why data science projects succeed or fail to deliver long-term value [112, 135, 11]. Consequently, data scientists can not only increase their output but also significantly improve the quality and real-world impact of their work.

**Empowering Domain Experts.** TFM-equipped agents offer domain experts an accessible interface, potentially through natural language interactions or guided workflows, to directly engage in data analysis themselves [93]. Domain experts can directly contribute deep contextual knowledge and ensure practical alignment. To this end, they have to clearly articulate the problem, validate the agent's understanding & approach against their domain-specific knowledge, and critically assess whether the results align with real-world objectives and constraints. Elevating domain experts to direct collaborators minimizes divergence from project requirements and physical realities that are common in traditional settings where data scientists serve as intermediaries [135, 64]. Additionally, collaboration with TFM-equipped agents democratizes data analysis, enabling domain experts to perform limited data analysis work themselves, even in organizations without dedicated data scientists.

## 4.2 Operationalizing the Data Science workflow with TFM-equipped agents

**We envision that workflows are *dynamically* orchestrated across abstract semantic operations** in data science processes, from problem formulation and data retrieval, to result validation. This allows for flexibility and adaptability across diverse tasks and domains, while enabling the optimization of individual operations and the deployment of context-dependent guardrails for limiting risks. Informed by discussions with data science experts, we have distilled a potential set of data science operations for these systems as illustrated in Figure 1, in line with preliminary systems [63, 50, 98] but with more explicit integration of human experts.

**The level of autonomy of the agent may vary based on the specific operation and use case**. For example, the retrieval of relevant data (tables, documents, etc.) requires knowledge of the domain context and exploration of heterogeneous data sources [80], necessitating agents with a high level of autonomy, hence requiring a less constrained action space. In contrast, operations and use cases with higher risks require strong guidelines and involvement of human expertise. An example of a high-risk operation is the selection of the analysis tool or TFM for a task and dataset at hand. Benchmarks have shown the tendency of LLMs to take shortcuts and skip human input where needed in tabular data analysis tasks [138, 116]. For instance in [84], an LLM sneakily guesses a causal relationship from parametric knowledge instead of using the provided data.

To avoid this, agents with high-risk tasks should be **constrained with a specified tool and mapping and verifiable and augmentable by humans**. Dedicated primitives should explicitly handle operations like causal inference or bias checking during data preparation stages, thus enforcing methodological rigor and consistency. Similarly, humans can be integrated by mapping their interactions to certain primitives that describe how and when human experts should be involved (e.g., to disambiguate the meaning of data). Besides constraining the agent's action-space and thus enable better and more constrained reasoning, human data scientists as well as domain experts are key in evaluating the validity and appropriateness of the tool and model selection for the insight needs at hand. Yet, despite many calls for human oversight, there is still no systematic design pattern that specifies *when* or *how* the human is pulled back into the workflow.

## 5 Risks to Navigate

The integration of LLM agents into end-to-end data science workflows raises significant technical, ethical, and operational concerns. As these agents increasingly automate tasks such as data processing, analysis, and even code execution, their limitations and failure modes must be critically examined.

**Expertise and Automation Bias.** LLM agents risk marginalizing human experts through automation bias, the tendency to over-trust model outputs [45]. Recent evidence shows AI assistants can be "detrimental to human skill learning and expertise" [87], as automated systems reduce opportunities for practice and skill development. This is already happening, predictions of radiology automation led to decreased training investments despite the continued need for human expertise [100]. Additionally, the human-AI gap creates risks in both directions: domain experts without statistical training may misuse AI tools, while data scientists may miss critical domain context.

**Model Misuse and Data Bias.** Data processing workflows are susceptible to subtle but critical errors such as data leakage [71, 70, 46]. Although there is no direct evidence that LLM agents are more prone to these errors, a lack of knowledge of the provenance of the data and the compartmentalization of agentic workflows might increase the associated risks. The ethical and statistical hazards of using AI systems trained on biased or poorly understood datasets raise concerns about social discrimination and data misuse [33], particularly without domain-informed oversight.

**Security and Execution Risks.** Code-generating agents that execute SQL or Python pose significant safety risks. LLMs have been shown to generate vulnerable or harmful code even under seemingly benign prompts [97], which could be made much worse through new techniques such as training data poisoning [7]. The use of sandboxed execution environments and access management systems, adversarial testing [15], and specific delimitations of sensitive operations 4 are essential.

**Privacy, Memorization.** LLMs are known to memorize rare or sensitive data from their training corpora [53], which can be extracted with targeted prompting [18]. This raises serious concerns about compliance with privacy regulations such as GDPR and CCPA.[1] An agent who decides to train a predictive model on private data might then lead to unexpected private data leakage.

**Behavioral Pathologies.** Reward hacking and sycophantic behavior has been regularly observed in reinforcement learning based agents [117, 99, 75, 8]. In data science, such behavior could lead to hard-to-detect test set leakage, multiple comparison gaming to generate fake insights, or suggesting complex and wasteful models to unsuspecting users.

**Sustainability and Resource Cost.** Finally, agentic workflows typically iterate trials and errors, requiring extensive inference cycles. This markedly increases computation costs and energy consumption. For context, a frontier model such as o3 or DeepSeek-R1 [26] is estimated to consume over 33 Wh per long prompt [66], and agentic workflows multiply this footprint through repeated queries. As usage scales, the environmental impact of these systems becomes increasingly untenable. Required infrastructure lead to concentrated resources worrisome from a political economy standpoint [130].

## 6 Research Agenda and Call to Action

The convergence of agentic systems, tabular foundation models, and human expertise offers an unprecedented opportunity to fundamentally reshape data science. Yet, safely realizing this vision demands a targeted and interdisciplinary research agenda—one that acknowledges the unique needs of data-centric workflows, prioritizes meaningful human-agent collaboration, and reorients the field toward long-term value creation rather than superficial benchmark gains. We outline below the concrete research directions we believe are necessary to enable this transition.

### 6.1 Evolving the Core Pillars of Modern Data Science

**Agenda for Agents.** A lot of the issues currently hindering data-science agents stem from more general issues plaguing agents, including long horizon planning, very high reliability and error correction to stay on rails, better tool use integration etc. These issues are the object of intense focus in academia and AI companies, and we believe there will be rapid progress on them. As described in section 4, we also believe that designing new high-level primitives for data science would make these agentic workflows much more efficient and safe. To this end, research on building agents optimized for interacting with humans, and deferring to humans or asking for clarification in critical moments is key. This involves accurately modeling user goals and expert knowledge [120], ideally in transparent and configurable ways [20], and developing efficient strategies to elicit such information while minimizing the user's cognitive and time burden [94, 74].

---

[1] https://gdpr-info.eu and https://oag.ca.gov/privacy/ccpa

**Agenda for Tabular Foundation Models.** Previous work [128] already highlighted important research directions for predictive TFMs. While such models have improved quickly since then, we believe progress in two main areas is still essential: scalability and contextualization. Predictive TFMs such as TabPFN [61] and representation-focused models like TaPAs [59] are currently strictly constrained on the number of input data points and features, and incur high inference costs. To address these issues, several promising techniques can be adapted to the tabular setting, including architectural innovations like linear attention [140], retrieval-augmented methods and fine-tuning [126], prompt tuning approaches [36], or the use of hypernetworks [91]. Another severe constraint is that SOTA predictive TFMs like TabPFN [61] are purely statistical reasoners, without any contextual knowledge about the task, dataset or the semantics of columns and values. To close this gap, TFMs need to be able to ingest much more diverse inputs, in particular text descriptions, annotations of data and tasks, contextual meta-data, and expert knowledge, for instance in the form of elicited probabilistic or causal priors. Moving in this direction requires bridging the gap between predictive TFMs, often trained on synthetic data with strong numerical signals, and representation TFMs, trained on contextually rich data, potentially through training on more diverse tables [86] or knowledge graphs [72]. Both model types have complementary strengths to offer: predictive TFMs would benefit from improved contextual understanding, while representation TFMs would benefit from a better understanding of numerical values and patterns [62], as well as missing tabular specific biases [22].

## 6.2 Building and Evaluating End-to-End Systems

Building an end-to-end system for assisted data science poses many challenges beyond agents and TFMs. Scoping, goal setting, evaluation and verifiability are aspects of the full system that are not addressed by research on agents or TFM themselves.

**Beyond Prediction.** Data science is more than data preparation and predictive modeling. Data science can be split in three groups of tasks [58]: (a) exploration, description, and hypothesis generation, i.e., getting to know your data and formulating what the questions of interest are, (b) answering causal questions about effects in the data, and (c) predictive tasks, usually as part of an automated decision-making process. A critical part of each of these is formulating the problem, identifying which tools are appropriate, and mapping the concepts required to apply the tool to the data. This crucial step is often assumed solved in machine learning benchmarks, which usually come with clearly defined tasks and metrics. The machine learning community so far has focused on selecting prediction methods when given a clear task, mapping the data schema to the task, and building ample benchmarks for this task [48, 43]. While there are definitions for insight generation [29, 98], there is no accepted definition or metric for Exploratory Data Analysis tasks, though empirical benchmarks exist [54]. Benchmarks for causal inference usually assume well-phrased and grounded questions and are typically limited in scope. Mapping to the correct task and validating the choice of method has received little to no attention in all areas. With this broadened scope, it is usually impossible to simply rank algorithms based on their performance given the data, as the "correct" answer might be highly context-dependent and potentially unknowable.

**More Realistic Evaluations.** Current tabular benchmarks have significant discrepancies with real-life settings. They focus on well-framed problems, and often neglect important aspects like distribution shift across time [110], data-specific data preprocessing [110, 127], or finding the relevant data [17]. To evaluate agentic systems, benchmarks containing tables with varying levels of human preprocessing are important to understand the performance of the entire pipeline. In practice, data is usually part of multiple tables in a larger database schema [31], and selecting and aggregating the correct data is an everyday task for most data scientists. However, only a few benchmarks reflect this reality today [109, 141, 68]. This relational multi-table setup could be addressed within TFMs themselves or by the integration of TFMs with other agent tools like Text-to-SQL [78]. Furthermore, benchmarks should measure performance on realistic context-rich tables on which a mix of agents and TFMs can shine by retrieving useful information from parametric knowledge, databases, or knowledge graphs [141, 68, 67]. Adding human interactions into the workflow further complicates evaluation. New evaluation metrics and evaluation protocols, including varying degrees of human interaction are needed. Such human-in-the-loop evaluations have been instrumental in biosecurity evaluating risks [96], and have been the gold standard of visualization and interpretability research [107, 137]. For tabular data analysis, the CollaborativeGym benchmark [116] provides a compelling starting point for such benchmarks, but work on much broader tasks is necessary.

**Embracing a New Flexibility of Goals.** TFMs based on the principles of PFNs [92] enable addressing previously separate tasks in a unified framework. Access to the causal mechanism during pretraining allows for new training objectives that are more aligned with practical requirements and can be easily adjusted to suit new scenarios. An example of exploring this newfound flexibility are the drift-resilient TabPFN [57] which is robust to distribution shifts by using the known causal model during pretraining, and FairPFN [108] which addresses counterfactual fairness. AVICI [85] goes even further and infers the causal graph in-context, allowing for complex inferences. These are just the starting points for training models to answer more general inference questions, including causal ones, even outside of the usual i.i.d. framework. TFMs are also able to produce models of a given architecture using in-context learning [91] focusing on interpretable model families, such as Generalized Additive Models [90]. These constrained architectures enforce white-box prediction functions and could extend to domain-specific families meaningful to experts. While TFMs can widen the range of scenarios we can address computationally, agents can help in selecting appropriate methods, validate assumptions, and guard against common mistakes, such as target leakage or non-i.i.d. data. Without the semantic understanding provided by LLM-based agentic systems, numerical tools alone can not detect these failure cases [58].

**End-to-end Verifiability.** To ensure trustworthy and aligned outcomes any computation and reasoning done through TFM or agent needs to be fully transparent to the human user. Only when human experts can review the data, context, and reasoning used for a conclusion can they fully trust and build upon the results. While there is a long line of work on data provenance [47, 44, 21, 102, 46] and LLM reasoning [5, 23, 10], as well as a growing interest in interpreting TFMs [111], combining these to produce explanations of end-to-end systems that are faithful and verifiable is an open challenge.

**Deployment.** Many challenges in data science projects only arise during deployment when transitioning from the well-defined development environment to the brittle physical world [112, 135]. Therefore, the collaborative framework should extend to deployment, accounting for monitoring deployed models, detecting drifts, and model updates. Novel capabilities, such as the ability of agentic systems to re-use pre-trained TFMs via in-context learning in different applications may open up opportunities to simplify deployment infrastructure.

## 7 Alternative Views

**Alternative view 1:** *LLM-powered agentic systems will be enough. There are so many people working on LLMs, LLMs will learn to reason better than TFMs soon, rendering TFMs unnecessary.*
**Rebuttal:** LLM architectures designed for sequences are inherently misaligned with the concept of columns in a table; thus, standard LLMs cannot be as efficient for tables as the specially adapted architectures in TFMs. Concepts from LLMs will be extremely important, but adapted for TFMs.

**Alternative view 2:** *We should entirely automate data science. This is already a reality [49, 125].*
**Rebuttal:** These works demonstrate that it is feasible to build fully automated *predictive* data science agents; yet, we argue that this risks jeopardizing the integrity and quality of results in critical tasks and may thus outweigh their benefits. Hence, we advocate to supercharge humans who can bypass the tedious steps and focus on reviewing and validating the steps taken. Also, data science is much more than prediction, and the remaining tasks are much harder to automate fully.

## 8 Conclusion

The future of data science lies in a collaborative paradigm that integrates Tabular Foundation Models, agentic systems, and human expertise. TFMs provide structure-aware statistical reasoning, agents enable orchestration and accessibility, and humans ensure contextual understanding and critical oversight. This synergy can overcome current workflow fragmentation and unlock greater efficiency and quality, but must be guided carefully to avoid risks like automation bias and model misuse. We call for research into scalable, context-aware TFMs, robust human-agent collaboration, realistic benchmarks, and transparent, verifiable systems. Rather than replacing data scientists, these tools should amplify their impact.

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
