# OpenReview forum: "Unlocking the Full Potential of Data Science Requires Tabular Foundation Models, Agents, and Humans"
_NeurIPS.cc/2025/Position_Paper_Track — Submitted to NeurIPS 2025 Position Paper Track_

### Official Review · Reviewer_nqC5 · 2025-08-12

[review text omitted: it was posted to a different submission]

---

### Official Review · Reviewer_gKvC · 2025-08-15

**Significance:** 2
**Presentation:** 3
**Rating:** 5
**Confidence:** 3

**Summary:**

The paper argues that end-to-end, human-absent automation is the wrong target for tabular data science. Instead, it proposes a collaborative paradigm that tightly integrates tabular foundation models (TFMs), LLM-based agents, and human experts. It surveys the limits of TFMs (narrow task coverage), agents (insufficient rigor on tables), and humans (limited bandwidth), and contends that only their combination can deliver robust, explainable, and scalable data science. The authors sketch a workflow (problem formulation → data ops → modeling → validation) with explicit human checkpoints, discuss risks (automation bias, leakage, privacy, security, sustainability), and outline a research agenda (stronger TFMs, safer/structured agents, realistic benchmarks, end-to-end verifiability, deployment).

**Strengths:**

1.	Timely and coherent position. The triad (TFMs + agents + humans) is clearly motivated and well argued for tabular work where provenance, rigor, and context matter.
	2.	Well-structured survey + design blueprint. The paper crisply contrasts capabilities/limits (e.g., TFMs’ structure awareness vs. agents’ tool use vs. human judgment) and offers a workflow with clear human intervention points; this is useful to practitioners and researchers alike.
	3.	Balanced treatment of risks. The section on automation bias, leakage, privacy/memorization, execution risk, and compute/energy cost shows practical awareness beyond benchmarks.

**Weaknesses:**

1.	As a position piece it’s acceptable, but even a small case study (e.g., TFM-equipped agent on a realistic, multi-table task) would substantiate the claims and clarify trade-offs in accuracy, latency, and cost.
	2.	The paper could more sharply differentiate its blueprint from AutoML, program-aided LMs, graph-RAG, DSPy-style pipelines, and existing data-science agents (what’s truly new beyond integration + emphasis on tabular?).
	3.	How to choose between TFMs vs. LLM tools, how to constrain agent action spaces in “high-risk” steps, how to select/validate structures and resolve conflicts (e.g., contradictory intermediate results) remain underspecified.
	4.	The desiderata (explainability, completeness, etc.) lack concrete metrics/protocols (e.g., trace coverage/faithfulness, cross-table consistency checks, cost/energy accounting, human-in-the-loop efficacy).
	5.	The paper argues humans are indispensable, but provides few patterns for when and how to re-insert experts (triggers, escalation criteria, UI/UX primitives), or how to elicit and encode institutional knowledge.

**Questions:**

•	What are the minimal new mechanisms that distinguish your system from RAG + schema-enforced extraction + tool-calling agents?
	•	How is structure/tool selection automated? When does the agent invoke a TFM vs. write code vs. defer to a human?
	•	What guardrails do you envision for causal/biased analyses (e.g., primitives that make target-leakage or non-IID assumptions explicit and checkable)?
	•	Can you share a cost/latency analysis for multi-stage, human-in-the-loop pipelines, and any caching/reuse strategies?
	•	What is the verifiability artifact (e.g., provenance graph + scored checkpoints) you expect reviewers/users to inspect?

**Alternative Position:**

Yes, and alternative positions are well-considered and named but not addressed

**Author Identification:**

No.

**Context:**

3

**Discussion:**

3

**Ethics:**

["NO or VERY MINOR ethics concerns only"]

**Position:**

Yes, the paper argues for or against a position related to machine learning.

**Support:**

2

**Thoroughness:**

2

---

### Official Review · Reviewer_Fnmh · 2025-08-19

**Significance:** 2
**Presentation:** 3
**Rating:** 4
**Confidence:** 3

**Summary:**

The paper argues that the future of data science lies in collaborative new systems that integrate human experts, agents and tabular foundational models instead of autonomous data science agents. The authors compare the strengths of human experts with TFMs and LLM agents, and call for research in scalable, context-aware TFMs, human-agent collaboration, realistic benchmarks, and verifiable systems.

**Strengths:**

1. The paper is quite well-written and easy to understand.
2. It is also well-structured: to support their position, the authors first lay out current challenges in data science, then discuss the strengths and limitations in tabular foundational models, LLM agents and human experts. The paper then suggests the collaborative paradigm, potential risks and future research directions.
3. The authors also discussed with details and examples on the unique capabilities of human experts - domain knowledge that are not easily accessible for model training, contextual understanding of data science, and alignment with the final goal of doing data science.
4. The authors discussed thoroughly on the potential risks in AI centered data science.

**Weaknesses:**

The paper can benefit from the following discussions:
1. The paper's core position, that AI needs human experts for goal alignment and domain knowledge, doesn't seem particularly novel. It's described as a "deeply integrated" collaboration, but the workflow presented just looks like the standard process of humans formulating problems and then validating the AI's solutions in each step.
2. The argument for why a tabular foundational model is the right choice isn't backed by strong empirical or theoretical evidence. The reasoning rests on general LLM limitations like hallucinations and poor reasoning, but this overlooks the incredibly rapid progress made to solve these deficiencies in frontier models.
3. A major piece of the data science process, data ETL and processing, is missing from the discussion. This seems like a critical omission, especially since LLMs are promising for automating these tasks.

**Questions:**

1. I wonder if it would be more effective to discuss the three layers of data science as human supervision, agents, and models? We've seen from frontier lab demos that AI agents can now use tools like Python to create simple data science models or use existing models for problems that don't involve TFMs, maybe instead of arguing for tabular foundational models, just general machine learning models for data science?
2. The paper mentions a “dynamically orchestrated workflow,” but it’s not clear what this means concretely. Could you elaborate on what this looks like in practice? The figure that illustrates the workflow with AI and human seems like a typical work process with human review in the loop.

**Alternative Position:**

Yes, and alternative positions are well-considered and named but not addressed

**Author Identification:**

No.

**Context:**

3

**Discussion:**

2

**Ethics:**

["NO or VERY MINOR ethics concerns only"]

**Position:**

Yes, the paper argues for or against a position related to machine learning.

**Support:**

2

**Thoroughness:**

5

---

### Note · Authors · 2025-09-05

**1-10 Additional Comments:**

It was a bit unclear to us where the discussion about the position should actually take place. For example, it could make sense to have a “managed” discussion phase during the review process on OpenReview that is then also used to determine which papers get accepted.

**1-11 Submit Again:**

Probably yes

**1-1 Submission Process:**

4

**1-3 Future Development:**

It would be interesting to facilitate the "discussion" of the presented positions with the NeurIPS audience, e.g., through discussion groups or panels.

**1-4 Interest:**

["Structured debates on controversial topics"]

**1-5 Thoughtful:**

6

**1-6 Supportive:**

5

**1-7 Technical Aspects Versus Position:**

6

**1-8 Gate Keeping:**

10

**1-9 Camera Ready Changes:**

For the camera ready we add a case-study to illustrate the failures and missed opportunities resulting from the current disconnect between human experts and LLM agents on tabular data [nqC5, gKvC]. To ground this, we highlight real-world examples like the failure of Zillow's iBuying business [1], where an intense focus on data and modeling work hindered an effective response to a market drift not captured in historical data. We also draw from literature showing that agents alone can succeed on well-scoped problems (e.g.,
Kaggle competitions [4]), but falter in context-rich domains like healthcare, where hallucinations are common [2] and results on clinical tasks are unreliable [3].

Additionally, we clarify the changing role of humans in data science by preparing a section outlining an “Agenda for Data Scientists and Domain Experts” to chapter 6.1 [gKvC, Fnmh]. This agenda formulates the need for humans as orchestrators that constantly update and configure systems when data or business circumstances change, as experts that contextualize the process in organizational and project related realities, and as reviewers that control the process and account for critical scenarios. In line with this we also rework Figure 1 to better reflect the involvement of experts.

In addition, we make smaller writing improvements that clarify our argumentation, particularly strengthening the argument for TFMs [gKvC, Fnmh]. We integrate the need for new evaluation protocols that account for collaborative data science into Section 6.2 [nqC5], and also rename section 3.3 to “Humans are experts, but don’t scale” to align better with the other subchapters of chapter 3.

[1] “In Defense of Zillow’s Besieged Data Scientists”, https://bit.ly/4p81F6q
[2] ”Google’s healthcare AI made up a body part”, https://bit.ly/47m2CSe
[3] Jiang et al., 2025, https://arxiv.org/abs/2501.14654
[4] Grosnit et al., 2024, https://arxiv.org/abs/2411.03562

**3-1 Review Response1:**

Fnmh

**3-2 Reaction To Review1:**

The review challenges our position, taking a clearly critical stance towards the necessity and role of human experts and TFMs. We want to highlight that this paper reflects a position from key researchers in the involved research areas supporting the important message to steer away from the LLM-centric path for tabular data science, which will not utilize its full potential and is prone to dangerous risks. We clarify some of our arguments below.


While we agree with the reviewer that problem formulation and continuous validation are key tasks for human experts, we argue that their most critical role lies in contextualizing the data science process in organizational and project related realities, steering the process in alignment with the overarching goals, and accounting for critical scenarios. This is a dynamic collaboration, not just supervision. To better illustrate this, we add an "Agenda for Human Experts" and revise Figure 1 to show how humans actively guide the process.


While we share the reviewer's sentiment of the progress in LLMs, improving both generally and on tabular-specific challenges (e.g., numeracy, longer context), we argue that these developments widen the gap between LLMs and TFMs, as advances in LLMs often directly benefit TFMs as well (e.g., better optimization and robust inference). We also argue that TFMs have distinct advantages over “general” ML models trained by an agent. For example, TFMs allow for easier and more robust deployment, allowing organizations without that expertise to scale their data science capabilities. TFMs can also support a much richer set of inputs (e.g., user priors and dataset context) and outputs (insights about the data, conditional predictions…), significantly increasing the bandwidth between agents, humans, and TFMs, enabling the synergy we’re arguing for.


Overall, this review highlights that our position paper stimulates fundamental and novel discussions and is of interest to the NeurIPS community.

**3-3 Review Response2:**

gKvC

**3-4 Reaction To Review2:**

The review is clearly supportive of our position, brings up thoughtful technical suggestions that help us improve the camera ready version of the paper, and illustrates how the position gives rise to important new research questions. We have taken up the idea of adding a small case study, through which we further concretize the details of our position with pointers to existing real-world references (e.g., when to choose TFMs vs. LLMs) and fleshed out example cases (e.g., when existing agent-based systems work and when they fail), addressing W1-W5. For example, existing agentic systems can succeed on well-scoped tasks like those on Kaggle. However, they struggle in less structured, high-stakes domains like healthcare where context is sparse and errors like hallucinations or omissions can be dangerous.


To address the additional questions of the review, we also add an “Agenda for Humans Experts” to the paper, which reflects our intention to redirect the research community in the direction of collaborative data science. We argue for a new framework built on a dynamic, collaborative interplay between agents, TFMs, and human experts, which replaces the linear prompt-and-execute workflow and elevates humans from users to integral partners.


The reviewer also poses some interesting questions that further highlight why researching the interplay of humans, agents, and TFMs should be a priority, while it is also a key motivation for our position paper to pose and prioritize such questions in future research (Q2-Q5). We extend our discussion to include these points (e.g., regarding high-risk scenarios where humans must be able to restore a “clean” state) and point towards potential solutions (e.g., version control mechanisms for agents). Fully answering some of the questions (e.g. cost/latency analysis) requires extensive experimentation to draw empirical conclusions, which is beyond the scope of this position paper.

**3-5 Review Response3:**

nqC5

**3-6 Reaction To Review3:**

The review is clearly supportive of our paper. It also makes some helpful suggestions, such as the inclusion of concrete case studies to strengthen feasibility claims. We have taken up this idea, which also allows us to further concretize how our suggested combination of LLMs, TFMs, and humans could play out based on fleshed-out example cases. We elaborate on this revision item in the summary of Camera Ready changes.


The review also highlights the importance of new evaluation protocols and benchmarks that are specifically designed for collaborative data science systems, which are currently missing in this field. We will emphasize the need for new evaluation protocols that account for collaborative data science systems in Section 6.2.

---

### Meta-Review · Area_Chair_Lkrv · 2025-09-18

**Rating:** 5
**Confidence:** 3

**Strengths:**

Good engagement with prior literature.
Rich discussion of research agenda.
Well written.
Highlights why researching the interplay of humans, agents, and TFMs should be a priority.
Should get good engagement.

**Weaknesses:**

The paper's core position, that AI needs human experts for goal alignment and domain knowledge, doesn't seem particularly novel.
A major piece of the data science process, data ETL and processing, is missing.

**Questions:**

The paper could more sharply differentiate its blueprint from AutoML, program-aided LMs, graph-RAG, DSPy-style pipelines, and existing data-science agents.  How do these complement or affect your position.

**Thoroughness:**

2

---

### Decision · Program_Chairs · 2025-09-26

Reject